# Intraperitoneally Administered Vancomycin in Patients with Peritoneal Dialysis-Associated Peritonitis: Population Pharmacokinetics and Dosing Implications

**DOI:** 10.3390/pharmaceutics15051394

**Published:** 2023-05-02

**Authors:** Jan Miroslav Hartinger, Danica Michaličková, Eliška Dvořáčková, Karolína Hronová, Elke H. J. Krekels, Barbora Szonowská, Vladimíra Bednářová, Hana Benáková, Gabriela Kroneislová, Jan Závora, Vladimír Tesař, Ondřej Slanař

**Affiliations:** 1Department of Pharmacology, First Faculty of Medicine, Charles University and General University Hospital in Prague, 128 00 Prague, Czech Republic; danica.michalickova@lf1.cuni.cz (D.M.); eliska.dvorackova@vfn.cz (E.D.); karolina.hronova3@vfn.cz (K.H.); ondrej.slanar@lf1.cuni.cz (O.S.); 2Division of Systems Pharmacology and Pharmacy, Leiden Academic Centre for Drug Research, Leiden University, 2311 EZ Leiden, The Netherlands; e.krekels@lacdr.leidenuniv.nl; 3Internal Department of Strahov, General University Hospital in Prague, 128 00 Prague, Czech Republic; bszonowska@gmail.com; 4Department of Nephrology, First Faculty of Medicine, Charles University and General University Hospital in Prague, 128 00 Prague, Czech Republic; vladimira.bednarova@vfn.cz (V.B.); vladimir.tesar@vfn.cz (V.T.); 5Institute of Medical Biochemistry and Laboratory Diagnostics, First Faculty of Medicine, Charles University and General University Hospital in Prague, 128 00 Prague, Czech Republic; hana.benakova@vfn.cz (H.B.); gabriela.kroneislova@vfn.cz (G.K.); jan.zavora@vfn.cz (J.Z.)

**Keywords:** glycopeptides, therapeutic drug monitoring, methicillin resistant *Staphylococcus aureus* (MRSA), area under the curve (AUC), drug-exposure, renal replacement therapy, continuous ambulatory peritoneal dialysis, infection

## Abstract

Peritonitis is a limiting complication of peritoneal dialysis, which is treated by intraperitoneal administration of antibiotics. Various dosing strategies are recommended for intraperitoneally administered vancomycin, which leads to large differences in intraperitoneal vancomycin exposure. Based on data from therapeutic drug monitoring, we developed the first-ever population pharmacokinetic model for intraperitoneally administered vancomycin to evaluate intraperitoneal and plasma exposure after dosing schedules recommended by the International Society for Peritoneal Dialysis. According to our model, currently recommended dosing schedules lead to possible underdosing of a large proportion of patients. To prevent this, we suggest avoiding intermittent intraperitoneal vancomycin administration, and for the continuous dosing regimen, we suggest a loading dose of 20 mg/kg followed by maintenance doses of 50 mg/L in each dwell to improve the intraperitoneal exposure. Vancomycin plasma level measurement on the fifth day of treatment with subsequent dose adjustment would prevent it from reaching toxic levels in the few patients who are susceptible to overdose.

## 1. Introduction

Peritoneal dialysis (PD) is a modality of renal replacement therapy (RRT). Worldwide, approximately 11% of all patients undergoing dialysis are treated by PD. PD is performed by the instillation of hyperosmotic peritoneal dialysate fluid (either glucose, icodextrine, or amino-acid based) into the peritoneal cavity via a peritoneal catheter. PD fluid dwells there for a period of time to allow water, as well as soluble waste compounds and potassium, to diffuse from the blood to the peritoneal cavity. At the end of the dwell, the fluid is drained and discharged. The dwells are repeated several times a day in various and individualized schedules. 

The most common and limiting complication of PD is bacterial peritonitis [1,2]. International Society for Peritoneal Dialysis (ISPD) guidelines recommend vancomycin as one of the first-line antibiotics for the treatment of G+, mixed, or cultivation-negative peritonitis. As peritonitis associated with PD is increasingly caused by beta-lactam-resistant staphylococci and enterococci, the use of vancomycin is becoming an increasingly important treatment modality [2]. Vankomycin, as well as gentamicin, cephalosporins, and other antibiotics, should be preferably administered intraperitoneally (i. p.) diluted directly in the instilled peritoneal dialysate if the patient is not septic. During the treatment of peritonitis, the individual regimen is usually switched to the scheduled one with 4–5 manual exchanges per day (continuous ambulatory peritoneal dialysis, CAPD). The ISPD recommends vancomycin dosing for patients treated by CAPD of either 15–30 mg/kg once every 5–7 days (intermittent dosing) or 20–25 mg/kg loading dose followed by maintenance dosing of 25 mg per liter of infused peritoneal dialysate in each subsequent exchange (continuous dosing) [3]. However, scientific support for these regimens is limited, and target plasma or peritoneal levels remain unknown. Only a few clinical studies have studied plasma but not peritoneal exposure of vancomycin. 

Reported vancomycin absorption from the peritoneal cavity is 70–91% of the administered dose in patients with peritonitis [4,5], and this decreases in healthy subjects without inflammatory changes in the peritoneal cavity [4,5,6,7]. On the other hand, the diffusion of vancomycin from the blood into the peritoneal dialysate occurs when the drug concentrations in plasma are higher than in the peritoneal cavity [5,6]. The systemic pharmacokinetics of vancomycin in patients with end-stage renal disease (ESRD) is markedly altered due to a diminished clearance (CL) [8,9]. Nevertheless, systemic CL of vancomycin persists to some extent in patients with residual kidney function compared to anuric patients [10]. This is also likely to have clinical relevance as it has been shown that the probability of treatment success is lower when treating G+ or culture-negative peritonitis in patients with residual renal function compared to completely anuric patients [11]. 

Vancomycin efficacy relies on achieving a sufficiently high exposure that is optimally expressed by the ratio of the drug’s 24 h area under the curve (AUC_24_) to the minimum inhibitory concentration (MIC) [12]. It is therefore expected that the achievement of sufficient intraperitoneal exposure to vancomycin is critical for local efficacy. In addition, vancomycin (and other antibiotics) may be less efficacious when diluted in peritoneal dialysis fluid. In the study by Tobudic et al., vancomycin completely lost efficacy against methicillin-resistant *Staphylococcus aureus* (MRSA) while only the bacteriostatic effect of the dialysate solution itself was observed [13]. The currently recommended AUC_24_:MIC ratio for vancomycin is ≥400, which corresponds to AUC_24_ of 400–600 mg×h/L in patients with infection caused by susceptible bacteria. Lower exposition may cause treatment failure, whereas higher exposures substantially increase the risk of systemic toxicity [12]. Systemic toxicity could be prevented by reaching high levels only locally in the peritoneum while maintaining low systemic exposures. This equilibrium should be achieved by appropriate i. p. dosing of vancomycin. Given that i. p. vancomycin administration would yield minimally fluctuating plasma concentrations, plasma target levels of 20–25 mg/L, similar to continuous i. v. administration [12,14], may be suitable. A robust vancomycin pharmacokinetic model and the measurement of vancomycin intraperitoneal levels at the end of the dwell time (i.e., in the drained-out dialysate) may be helpful to calculate the overall intraperitoneal exposure to vancomycin and tailor the treatment for an individual patient.

As resistance to beta-lactams is not rare and resistance to vancomycin is steadily increasing [2], undoubtedly also due to frequent under-dosing, our aim was to propose a reliable dosing schedule. For this purpose, we build a population pharmacokinetic model for vancomycin in ESRD patients treated with PD and suffering from peritonitis, based on plasma and intraperitoneal levels of vancomycin measured in two dialysis centers. We also measured the ex vivo efficacy of vancomycin and vancomycin + gentamicin diluted in the peritoneal dialysis fluid. Finally, we performed simulations to illustrate the implications our findings may have on the dosing of vancomycin in ESRD patients with peritonitis treated with PD.

## 2. Materials and Methods

### 2.1. Patients and Dosing

An open-label retrospective observational study was performed in all adult ESRD patients treated by PD with suspected peritonitis, admitted to one of the two nephrology departments of the General University Hospital in Prague between June 2016 and August 2022, who were treated with vancomycin and who had at least one vancomycin level in plasma or peritoneal dialysate measured. At admission, all patients switched to 4–5 exchanges per day (CAPD) for the treatment of peritonitis. The standard initial peritonitis treatment in our facility is cefazoline + gentamicin with a possible switch to more targeted therapy after cultures become available. Vancomycin is therefore reserved only for not-responding culture-negative or beta-lactam-resistant G+/mixed infections. In most of the patients, we followed the ISPD recommendations of a loading dose of 15–30 mg/kg followed by maintenance doses of 25 mg per liter of peritoneal dialysate in all subsequent exchanges [3]. The plasma levels have been monitored and maintenance doses adjusted to reach the target AUC_24_ for vancomycin plasma levels of 400–600 mg×h/L, recommended by Rybak et al. [12]. When systemic infection was present, i. v. dosing was combined with i. p. dosing, as needed. Data from 3 patients who were treated only by i. v. vancomycin with measured i. p. levels were also added to the database. The i. p. levels were available in cases of suspected underexposure during the treatment. Since the study involved only the analysis of routine clinical data from TDM protocols and patient documentation, and at admission to the hospital, the patients signed an approved general informed consent wherein they state, inter alia, that anonymous data can be used for research, study specific ethics approval was waived. The study was carried out in compliance with the ethical principles of the Helsinki Declaration. 

### 2.2. Vancomycin Analysis 

Vancomycin levels in plasma and dialysate samples were determined using a nephelometric method (Beckman Coulter, Indianapolis, IN, USA). The analytical range for plasma samples was 3.5–40 mg/L. When higher levels were detected, the samples were diluted to obtain measurable values that were subsequently multiplied according to the dilution factor. The method was validated for peritoneal dialysate samples by measuring a series of vancomycin concentrations from 10–250 mg/L in high and low glucose concentration solutions and in icodextrin solutions. It was shown that icodextrin solution did not interfere with the measurement, but that glucose solutions decreased the measured value by approximately 20% regardless of the glucose or vancomycin concentration (Appendix A). Therefore, the vancomycin intraperitoneal levels obtained from the laboratory were multiplied by a factor of 1.2885 obtained from the linear regression analysis of levels measured in validation tests.

### 2.3. Population PK Analysis

A population PK analysis was carried out using NONMEM version 7.4.0 (ICON Development Solutions, Ellicott City, MD, USA) [15] and PsN v3.4.2 [16] both running under Pirana 2.9.0 [17]. Modeling was performed using the first-order conditional estimation method with interaction (FOCE-I). R v4.2.2 was used for the visualization of the data and model diagnostics. Model development was performed in three steps: Development of structural and statistical model.

For the structural model, the peritoneal cavity was considered one compartment, in which the actual volume of peritoneal solution for each patient was used as the volume of distribution. One- and two-compartment models were tested to describe the distribution of vancomycin throughout the rest of the body. The exchange of vancomycin was assumed to occur only between the intraperitoneal compartment and the central (plasma) compartment. First-order CL of vancomycin from the central compartment was assumed.

Log-normally distributed inter-individual variability (IIV) terms with estimated variance were tested on each PK parameter, with the exception of the volume of the peritoneal compartment (V1). As differences in the permeability of the peritoneal membrane were expected in patients with recurrent and relapsing episodes of peritonitis, inter-occasion variability (IOV) on intercompartmental clearance (Q) between the intraperitoneal compartment and the central compartment was also tested. An occasion refers to a particular vancomycin treatment course that ended up with vancomycin withdrawal due to the cure of the infection, switch to a different antibiotic when ineffective, or the removal of the PD catheter. Proportional, additive, and combination error models were tested for the residual error models for both peritoneal and plasma concentrations. 

2.Covariate analysis.

The following variables were tested as covariates: Body weight (BW), lean body mass (LBM), serum level of urea, creatinine, albumin, potassium, C-reactive protein at the beginning of the peritonitis treatment (CRP), estimated glomerular filtration rate (eGFR, calculated by the CKD-EPI 2009 formula), age, and volume of residual diuresis were tested as continuous covariates.Preserved diuresis (yes = over 500 mL urine daily or no = less than 500 mL urine daily), type of peritoneal solution (low, medium, or high glucose content, or icodextrin based), and sex were tested as categorical covariates.

A stepwise covariate modeling procedure was performed. Continuous covariates were tested in linear and power functions. Categorical covariates were tested by estimating the parameter value for one category as a fraction of the parameter value for the other. 

BW, LBM, sex, age, and albumin were tested as covariates for the central (plasma) volume of distribution (V2); BW, LBM, serum levels of urea, creatinine, and albumin, eGFR, sex, preservation, and volume of residual diuresis and age were assessed as covariates for CL; sex, BW, CRP, potassium, and type of peritoneal solution were tested as covariates for intercompartmental clearance (Q). 

For model selection, a decrease in the objective function value (OFV) of more than 3.84 points between nested models (*p* < 0.05) was considered statistically significant, assuming a chi-distribution. Additional criteria for model selection were the physiological plausibility of the obtained parameter values, the absence of bias in goodness-of-fit (GOF) plots, and acceptable relative standard error of the estimates (RSE) of structural model parameters.

3.Validation of the final model.

A bootstrap analysis was performed to assess the stability of the model. In this procedure, 500 replicates of the original data were generated by sampling patients from the original dataset with replacement. The final model was fit to each of the 500 resampled datasets, after which the median and 95% confidence intervals (CI) obtained for each parameter were compared with the final parameter estimates in the final model. 

The predictive properties of the structural and statistical model were validated using normalized prediction distribution errors (NPDEs). For this, the dataset was simulated 1000 times, after which the observed concentrations were compared to the range of simulated values using the NPDE package developed for R [18].

### 2.4. Model-Derived Dosing Implications

To illustrate the implications of our findings, simulations were performed with the final population PK model, which included IIV and RUV in model parameters. One thousand simulations for a typical individual with BW = 75 kg and eGFR = 6.76 mL/min with preserved diuresis and oliguria were performed for dosing regimens recommended by the ISPD (25 mg/kg once every 5 and 7 days (intermittent dosing), and a 25 mg/kg loading dose followed by MD of 25 mg/L in each subsequent exchange (continuous dosing)) over 21 days (the recommended length for *Staphylococcus aureus* peritonitis treatment according to ISPD guidelines [3]). For patients with eGFR = 6.76 mL/min, 50 kg and 100 kg with oliguria and preserved diuresis simulations were performed only for continuous dosing, as this dosing regimen was mostly followed in our medical facility. Further on, we performed simulations with the new proposed dosing regimen, which may improve intraperitoneal vancomycin exposure without increasing plasmatic levels over the safe border.

### 2.5. Microbiological Testing

The microbiological evaluation of the bactericidal activity of the peritoneal dialysate with vancomycin was adapted from the study by Wise et al. [19]. The dialysate samples from two patients (one treated with vancomycin only and one treated with vancomycin and gentamicin) with measured vancomycin and gentamicin concentrations, were used. The samples were diluted with tryptic soy broth in dilution sequences 1:2 from 1- to 1024-fold. Strains of *Staphylococcus epidermidis* CNCTC 5212 (ATCC 12228), *Staphylococcus aureus* CNCTC 5480 (ATCC 29213), and *Enterococcus faecalis* CNCTC 5483 (ATCC 29212) were cultivated for 24 h in 37 °C on Columbia blood agar. Bacterial suspensions of 0.5 McF (108 CFU/mL) were subsequently prepared. These suspensions were further diluted in TSA broth in a ratio of 1:100. Furthermore, 100 µL of diluted samples were then transferred to every well in a 96-well plate and incubated for 24 h at 37 °C. Ten micrograms of broth in wells that were not cloudy were then transferred to the Columbia blood agar and incubated for 48 h at 37 °C. The first dilution was regarded as bactericidal if 99.9% of bacteria growth was suppressed.

## 3. Results

Data from 41 patients during 57 hospitalizations were included and 132 or 241 peritoneal or plasma vancomycin concentrations were obtained, respectively. On these occasions, patients were treated for suspected or culture-validated peritonitis, including recurrent and relapsing episodes. The characteristics of the patients included in this analysis are listed in Table 1.

### 3.1. Final Population PK Model

Observed vancomycin peritoneal and plasma concentrations were best described by a two-compartment model with log-normally distributed IIV on clearance CL, V2, and Q (Figure 1). V1 refers to the peritoneal cavity and V2 to the rest of the body. Proportional and combination residual error models provided the best description of residual variability for concentrations measured in the peritoneal compartment and plasma, respectively.

The addition of IOV on Q improved the model fit significantly (*p* < 0.001). However, IOV was found to be small relative to IIV, meaning that variability between patients was larger than variability within patients on different occasions. Figure 2 illustrates how IIV and IOV on Q compare to each other. The inclusion of preservation of normal diuresis as a binary covariate (i.e., yes/no) on CL also resulted in a statistically significant improvement of the model fit (*p* < 0.001). Further improvement of the model fit (*p* < 0.001) was seen when eGFR was included as a covariate on CL in a linear relationship for patients with preserved diuresis (i.e., diuresis > 500 mL/day). Finally, adding BW in a linear relationship as a covariate on V2 resulted in an improvement of the model fit (*p* < 0.05), although the RSE for V2p increased from 9% to 78% with the inclusion of this covariate. Despite this increased uncertainty in the estimated value V2, we decided to retain the covariate relationship in the model as it is of high clinical value given that current dosing guidelines are based on BW. After the inclusion of this covariate relationship, none of the other covariates were statistically significant. The final parameter estimates obtained with this model are presented in Table 2. In the final model, V1, CL, V2, and Q for a typical patient with a median BW of 75 kg and a median eGFR of 6.76 mL/min with oliguria (urine output < 500 mL) are 2 L (volume of exchange), 0.192 L/h (RSE = 17%), 23.6 L (78%), and 0.544 L/h (16%), respectively.

The precision of the estimated structural parameter values was acceptable, with RSE values generally being well below the limit of 50% with the exception of V2p. The results of the evaluation and validation procedures of the final model are provided in the Appendix A. The basic GOF plots in plasma (Appendix A) indicate that the model describes the observed data accurately, with only a small trend in conditional weighted residuals (CWRES) versus population-predicted concentrations. There was no bias in basic GOF plots for peritoneal concentrations (Appendix A), indicating an accurate description of these concentrations by the model.

With the exception of V2p, median parameter values obtained with the bootstrap procedure were within 10% of the values obtained in the final model fit, indicating that the model results are robust. For V2p, the bootstrap median still only deviates 15% from the final parameter estimate, with the final parameter value being well within the 95% bootstrap interval, which increases the certainty of the obtained estimate. 

The distribution of the NPDEs obtained with the model for plasma concentrations has a mean of 0.1027 and variance of 0.848, and neither of these values is significantly different from the expected values of 0 (*p* = 0.254) and 1 (*p* = 0.253), respectively. For peritoneal concentrations, the distribution of NPDEs has a mean of 0.1228 and variance of 0.8438, and these values did not deviate significantly from the expected values of 0 (*p* = 0.381) and 1 (*p* = 0.8438) either. These results indicate that for both endpoints, the model has a slight but not statistically significant over-prediction of the variability. This means simulations with the model lean to the conservative side, as the variability that is predicted is slightly higher than can be expected in real life. Plots of NPDE distributions in Appendix A further confirm that the model accurately predicts concentrations in both compartments, without any trends over time or over the observed concentration ranges.

The model findings regarding the changes of Q in recurrent peritonitis episodes are graphically illustrated in Figure 2. It is clearly visible that the permeability fluctuates most likely due to the different severity of peritoneal inflammation during different peritonitis occasions.

### 3.2. Model-Derived Dosing Implications

Figure 3 shows plasma and peritoneal levels and AUCs for vancomycin peritoneal concentrations for a typical individual with BW = 75 kg and eGFR = 6.76 mL/min with preserved diuresis and oliguria for all dosing regimens recommended by ISPD. Appendix A depicts simulations for continuous dosing for BW = 50 kg and 100 kg with eGFR = 6.76 mL/min with preserved diuresis and oliguria. AUC_24_ is depicted for the 21-day-long treatment (the recommended length for *Staphylococcus aureus* peritonitis treatment according to ISPD guidelines [3]).

The target exposure of >400 mg×h/L is not reached in the peritoneum for most of the days during intermittent dosing. Furthermore, the treatment may only be borderline efficacious when recommended continuous dosing is applied in patients with preserved diuresis. Based on the simulations, it is clearly seen that during continuous administration, most of the patients would not reach even the target therapeutic plasmatic level of 20–25 mg/L. This allows for an increase in the intraperitoneal maintenance dosing during the continuous dosing approach. On the other hand, a large proportion of patients is already overdosed after the second dose when intermittent dosing every 5 days was used in oliguric patients even though insufficient exposure was reached in the peritoneum. Based on the presented model, we proposed optimal dosing (LD = 20 mg/kg followed by MD = 50 mg/kg in each subsequent dwell) that would allow us to increase the intraperitoneal vancomycin exposure without overdosing patients with excessive plasma levels. Simulated exposure with this regimen is depicted in Figure 4.

As our proposed dosing may lead to a slight overdose in some (especially oliguric, Figure 4B,D,F) patients, we recommend checking the vancomycin plasma concentration no later than on the fifth day of therapy and adjusting the vancomycin dosing as needed to ensure safe therapy. For a 50 kg patient, we used an exchange volume of 1.5 L due to the smaller peritoneal cavity. With a dosing schedule with 2 L exchanges in a 50 kg patient, the differences in plasma and peritoneal vancomycin levels were negligible (Appendix A).

### 3.3. Bactericidal Efficacy of Vancomycin Diluted in the Peritoneal Dialysate

Peritoneal dialysate samples were obtained from two patients, which were used to evaluate the bactericidal activity. Patient 1 was treated by i. v. vancomycin only. The vancomycin plasma concentration on the first day of the sampling was 25.19 mg/L, which is at the upper limit of the range recommended for continuous infusion [12], but the vancomycin concentrations in the i. p. samples were well below this value, as can be seen in Table 3, which shows the vancomycin levels and first dilution where bacterial growth was detectable. In the samples of glucose-based peritoneal dialysate with vancomycin, the bactericidal effect was unmeasurable for *Enterococcus faecalis* and it was only borderline effective for *Staphylococcus epidermidis* and *Staphylococcus aureus*, even though all experimental strains were vancomycin sensitive. Samples from patient 2, who was treated with the standard ISPD-recommended dosing schedules of continuous i. p. vancomycin + intermittent i. p. gentamicin for mixed peritonitis, showed similar vancomycin concentrations at the time of drain out (except for one sample where 1000 mg of vancomycin was infused intraperitoneally). Nevertheless, the bactericidal effect was much more pronounced, which is likely due to the presence of gentamicin.

## 4. Discussion

This is the first study using a population pharmacokinetic approach to describe the pharmacokinetics of intraperitoneally administered vancomycin in patients treated with PD. Our analysis showed the preservation of diuresis and eGFR to be the main determinants of CL, and BW to be a significant covariate for Vd. Moreover, model-based simulations indicated that current dosing recommendations by ISPD do not reach the exposure targets in the peritoneal cavity in the majority of patients with preserved diuresis. We have therefore recommended a new dosing regimen for these patients. 

Frequent peritonitis is one of the limiting factors of peritoneal dialysis treatment. It may lead to damage to the peritoneal membrane and, consequently, to the need to discontinue this method of RRT. Therefore, a peritonitis episode should be treated as soon as possible and aggressively enough to limit the damage [3]. Both the choice of the right agent and its adequate dosing is crucial to bringing the disease under control. Our model is highly accurate in predicting intraperitoneal levels based on the dosing schedule, as demonstrated by the evaluation methods. The simulations clearly show that the intermittent vancomycin dosing recommended by the ISPD (i.e., 15–30 mg/kg every 5–7 days) may lead not only to intraperitoneal underexposure but also to possible systemic overexposure of the patient after the second and subsequent doses (Figure 3). The ISPD-recommended continuous dosing schedule with a 20–25 mg/kg loading dose and 25 mg/L maintenance doses leads to intraperitoneal underexposure in a remarkable number of patients with preserved diuresis and residual renal functions and overexposure for 1–2 days after LD administration in some patients when 25 mg/L LD is used (Figure 3). The only subgroup of patients with relatively reliable intraperitoneal exposure during the currently recommended dosing is the subgroup of oliguric patients with continuous vancomycin administration. 

We have also proven that clinically important intra-occasional variability during different episodes of peritonitis occurs in peritoneal membrane permeability for vancomycin (Figure 2), which makes the prediction of plasma levels more variable. This is likely due to the differences in the severity of peritoneal membrane inflammation during a particular peritonitis episode. This may also be the reason why trough plasma vancomycin concentrations were not proven to correlate with treatment effectivity in other studies [20,21], as they also poorly correlate with intraperitoneal levels. Plasma concentrations of vancomycin were best described by a one-compartment distribution model (the second compartment in our final model refers to the peritoneal cavity). This is supported by some studies, although there are studies that described two- and even three-compartment models to provide the best description of vancomycin disposition [22]. Due to differences in parameterization and covariate relationships, a direct comparison of findings between population PK studies using a non-linear mixed-effects approach is difficult. To allow a comparison of PK parameters, we calculated parameter values for the typical individual from our study with a BW of 75 kg and an eGFR of 6.76 mL/min (Figure 3). A drastic variation in estimated Vd values has been reported in previous studies, ranging from 29.9 L to 154 L [22,23,24]. Moreover, these studies have not identified BW as a significant covariate, whereas many others have. In respect of CL, there are few data regarding ESRD patients; in patients undergoing high-flux hemodialysis (HD), non-HD CL of vancomycin was 0.443 L/h [25]. 

Based on our model, we recommend using the lowest recommended intraperitoneal loading dose for a continuous dosing schedule of 20 mg/L and then administering maintenance doses of 50 mg/L for each dwell. This yields sufficient exposure in the peritoneum from the beginning of the treatment. Due to the variations of peritoneal membrane permeability in different peritonitis occasions, plasma levels are difficult to predict. Therefore, plasma levels should be monitored no later than on the fifth day of treatment to adjust the dosing in patients, who would be at risk of excessive systemic exposure and repeatedly thereafter as the levels may steadily grow in some patients due to the slow vancomycin accumulation during treatment in patients with a markedly prolonged elimination half-life. Typically, oliguric patients would likely need to reduce the dosing when reaching plasma levels over 25 mg/L.

Our recommended dosing will maximize peritoneal exposure where the AUC_24_ consistently exceeds the target values recommended by Rybak for systemic levels [12]. One may argue that the peritoneal levels could be unnecessarily high, but it is likely not the case as the efficacy of vancomycin in glucose and icodextrine-containing peritoneal dialysates is largely reduced as shown in our ex vivo samples (Table 3), as well as previously published data [13]. Therefore, intermittent vancomycin administration should not be used anymore, and the required AUC for intraperitoneal vancomycin during continuous dosing should be higher than the recommended values for plasma levels. At the same time, local high vancomycin exposure in the peritoneum without excessive plasma levels will not represent a risk for systemic toxicity. Therefore, we believe that vancomycin should be dosed to reach the highest possible i. p. exposition, which is only limited by systemic toxicity when plasma levels exceed the recommended value.

Our study has several drawbacks. All but one patient included in the study were Caucasians so we could not test the possible influence of ethnic origin on the pharmacokinetics of vancomycin. We also used commonly available markers of inflammation as CRP, which did not correlate with intercompartmental clearance, but we cannot exclude that more sensitive markers such as procalcitonin would show some correlation as they may more precisely correlate with peritoneal inflammation. We also measured the residual glomerular filtration rate from the creatinine level, which is less precise in comparison to cystatin C or other more precise methods.

## 5. Conclusions

We have developed a reliable population pharmacokinetics model for intraperitoneal vancomycin administration based on TDM data from 57 treatment occasions. Our PK model shows that the recommended ISPD dosing schedules for vancomycin may lead to under-exposure in a considerable number of patients when intraperitoneal rather than plasma concentrations are regarded as the treatment target. Based on simulations, we propose a novel dosing schedule for intraperitoneal vancomycin with a 20 mg/kg loading dose and 50 mg/L maintenance doses that should be followed by adjustment according to repeated TDM of plasma levels, beginning on day 5 at the latest. This dosing approach will allow the highest possible intraperitoneal exposure without reaching toxic plasma concentrations and therefore shall provide more efficacious treatment of peritonitis caused by beta-lactam-resistant G+ infections.

## Figures and Tables

**Figure 1 pharmaceutics-15-01394-f001:**
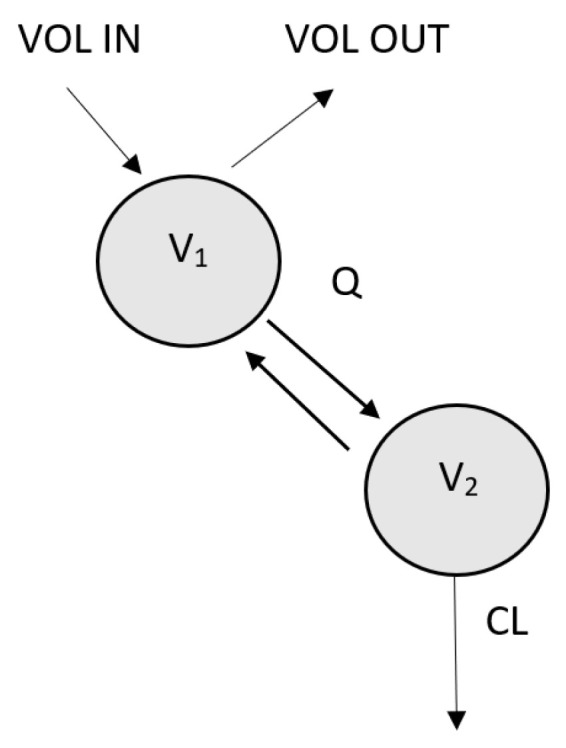
Schematic representation of the pharmacokinetic model for vancomycin in ESDR patients with peritonitis treated with peritoneal dialysis. V1 = volume of peritoneal compartment (peritoneal cavity), V2 = volume of central (systemic) compartment, CL = clearance, Q = intercompartmental clearance, VOL IN—volume of infused peritoneal dialysate, VOL OUT—volume of dialysate that was drained out at the end of the dwell.

**Figure 2 pharmaceutics-15-01394-f002:**
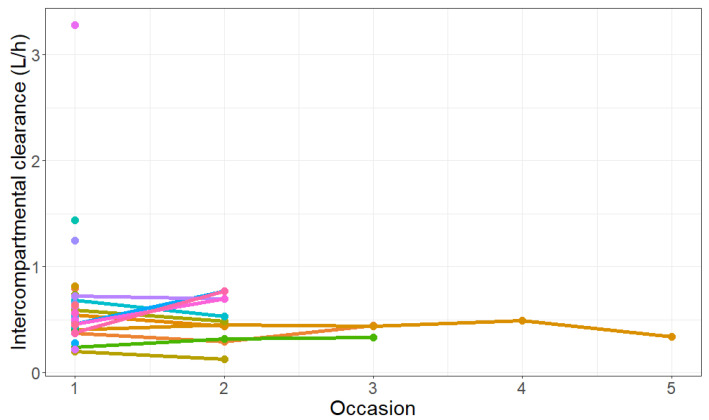
Inter-compartmental clearance values for all patients on different occasions. Different colors represent different subjects.

**Figure 3 pharmaceutics-15-01394-f003:**
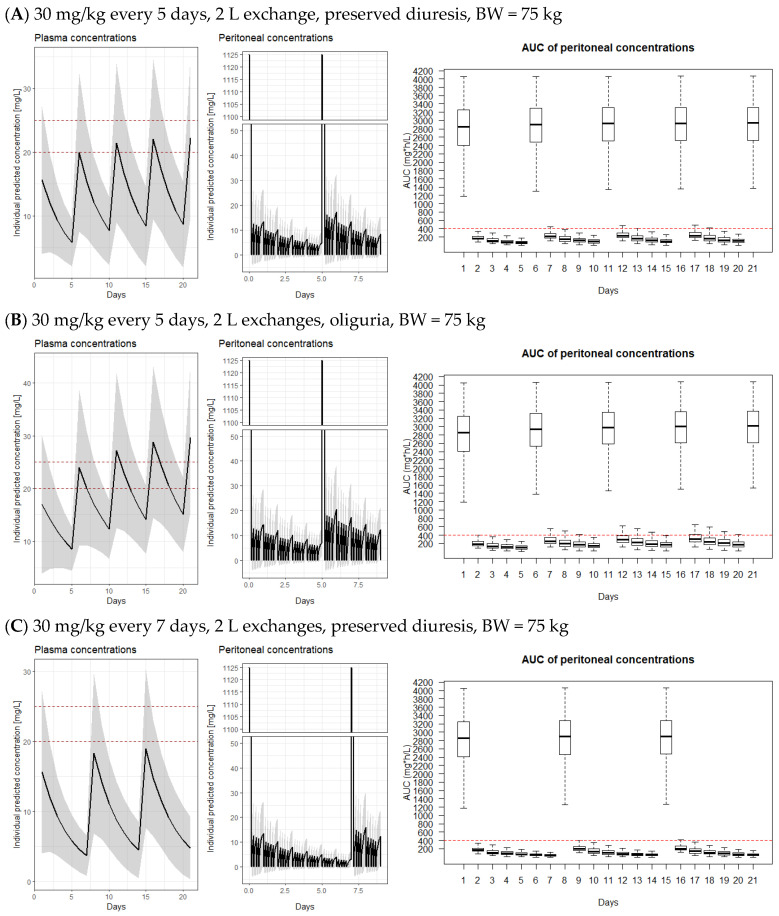
Simulations of vancomycin exposure in the peritoneal fluid and plasma upon recommended ISPD dosing schedules. AUC_24_ is calculated for a CAPD schedule with 5 daily exchanges (i.e., 4 × 4 h and 1 × 8 h) in a patient with BW = 75 kg. When preserved diuresis is stated, residual eGFR was set to 6.76 mL/min (median value of the model). Red lines in the left graphs—target plasma concentrations range for continuous vancomycin infusion, red line in the right graphs—target vancomycin AUC (i.e., 400 mg×h/L) that should be reached for effective therapy. AUC—area under the curve of peritoneal concentrations. LD—loading dose, MD—maintenance dose. Oliguria = less than 500 mL/day; Preserved diuresis = more than 500 mL/day. Solid lines represent median of simulated concentrations, shaded areas represent 95% confidence interval of simulated concentrations; box plots with whiskers from minimum to maximum are presented for AUC_24_ for 21-day-long treatment.

**Figure 4 pharmaceutics-15-01394-f004:**
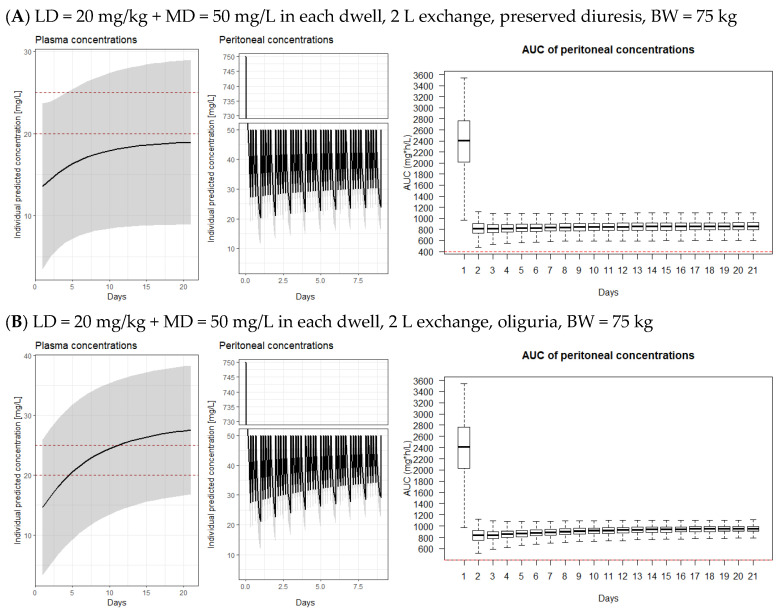
Expected vancomycin i. p. and plasmatic concentrations and exposure after proposed dosing. AUC_24_ is calculated for a CAPD schedule with 5 daily exchanges (i.e., 4 × 4 h and 1 × 8 h) in patients with BW = 50 kg, 75 kg, and 100 kg. When preserved diuresis is stated, residual eGFR was set to 6.76 mL/min (median value of the model). Red lines in the left graphs—target plasma concentrations range for continuous vancomycin infusion, red line in the right graphs—target vancomycin AUC (i.e., 400 mg×h/L) that should be reached for effective therapy. AUC—area under the curve of peritoneal concentrations. LD—loading dose, MD—maintenance dose. Oliguria = less than 500 mL/day; Preserved diuresis = more than 500 mL/day. Solid lines represent median of simulated concentrations, shaded areas represent 95% confidence interval of simulated concentrations; box plots with whiskers from minimum to maximum are presented for AUC_24_ for 21-day-long treatment.

**Table 1 pharmaceutics-15-01394-t001:** Characteristics of the included patient population.

Parameter (Unit)	Value *
Age (years)	68 (53–74)
Gender (F/M)	17/24
Body weight (kg)	75 (70–84)
Body mass index (BMI, kg/m^2^)	26.67 (21.6–28.34)
Lean body weight (LBW, kg)	54.31 (46.18–59.89)
Ideal Body Weight (IBW according to Devine equation, kg)	63.28 (59.28–72.18)
Fat Free Mass (FFM, kg)	54.3 (46.2–59.9)
Preserved diuresis (mL/day)	1000 (350–1450)
Oliguria (<500 mL/day), N (%)	10 (24.4%)
Preserved diuresis (>500 mL/day), N (%)	31 (75.6%)
Creatinine (µmol/L) **	694 (564–849)
Estimated glomerular filtration rate (mL/min/1.73 m^2^) ***	6.76 (5.07–7.92)
Urea (mmol/L) **	18 (14.9–22.3)
Uric acid (µmol/L) **	313 (269–349)
C-reactive protein (mg/L) **	31.7 (10.9–96.5)
Serum potassium (mmol/L) **	4.1 (3.8–4.6)
Serum albumin (g/L) **	28 (26–31)
Patients with 1500 mL exchanges, N (%)	16 (39%)
Patients with 2000 mL exchanges, N (%)	23 (56%)
Patients with 1000 mL exchanges, N (%)	1 (2%)
Patients with 1200 mL exchanges, N (%)	1 (2%)
Cultivations in peritoneal dialysate	
culture negative	13 (23%)
G+	36 (63%)
G−	4 (7%)
mixed	3 (5%)
G+, candida	1 (2%)
Concomitant exit-site infections	27 (47%)
Months on PD	22.5 (9–46.75)

* Values are presented as median (interquartile range) unless stated otherwise. ** Level from sample drawn at the beginning of the treatment with vancomycin. *** Calculated according to CKD-EPI 2009 formula, only in patients with preserved diuresis.

**Table 2 pharmaceutics-15-01394-t002:** Parameter estimates of the final population PK model of vancomycin in patients treated with CAPD.

Parameter [Units]	Final Model (RSE %)	Bootstrap Median (Range)
Fixed effects		
V1 [L]	FIXED *	
CLi [L/h] = CLp × ((1 + θRESDIU) × (CRCL/6.76))^RESDIU>500^		
CLp [L/h]	0.192 (17%)	0.186 (0.144–0.267)
θRESDIU	1.26 (31%)	1.36 (0.50–2.25)
V2i [L] = V2p + θBWV × (BW/75)		
V2p [L]	23.6 (78%)	27.2 (1.68–65.3)
θBWV	50.9 (38%)	45.9 (9.6–76.0)
Q [L/h]	0.544 (16%)	0.53 (0.40–0.76)
Inter-individual variability		
CL (%)	34.2% (22%)	32.2% (16.0–45.8%)
V2 (%)	30.3% (21%)	29.2% (13.8–40.4%)
Q (%)	50.7 (23%)	45.3% (20.1–79.6%)
IOC on Q (%)	31% (37%)	30.1% (17.3–51.7%)
Residual variability		
Proportional error, peritoneal concentration	0.091 (15%)	0.089 (0.064–0.146)
Additive error, plasma concentration	4.72 (31%)	4.74 (1.38–7.11)
Proportional error, plasma concentration	0.00604 (54%)	0.00649 (0.000450–0.01489)

* Fixed to the actual volume of peritoneal solution used (range 1–2 L); Abbreviations: RSE = relative standard error of the estimate; CLi = individual clearance value; CLp = population clearance value; eGFR = estimated glomerular filtration rate according to CKD-EPI (2009) equation; θRESDIU = increase in CL when residual diuresis (RESDIU) is preserved; RESDIU>500 = binary parameter indicating whether residual diuresis is preserved (1) or not (0); V1 = volume of peritoneal compartment; V2i = individual volume of central (systemic) compartment value; V2p = population volume of central (systemic) compartment value; θBWV = increase in V2 per kg bodyweight.

**Table 3 pharmaceutics-15-01394-t003:** The bactericidal efficacy of peritoneal dialysate samples from two patients.

**Patient 1—Intravenous Vancomycin, Infection of Unknown Origin, *Staphylococcus lugdunensis* in Exit-Site of PD Catheter; Plasma Level = 25.19 mg/L**						
Vancomycin concentration at the end of the dwell (mg/L)	6.67	8.14	11.78	7.78	8.03	12.19
Dialysis solution	glucose (1.36%)	glucose (2.27%)	glucose (2.27%)	glucose (2.27%)	glucose (1.36%)	glucose (1.36%)
*Staphylococcus epidermidis*	unmeasurable (<1:2)	1:2	1:2	1:2	1:2	unmeasurable (<1:2)
*Staphylococcus aureus*	unmeasurable (<1:2)	unmeasurable (<1:2)	1:2	1:2	1:2	1:2
*Enterococcus faecalis*	unmeasurable (<1:2)	unmeasurable (<1:2)	unmeasurable (<1:2)	unmeasurable (<1:2)	unmeasurable (<1:2)	unmeasurable (<1:2)
**Patient 2—intraperitoneal vancomycin and gentamicin; peritonitis caused by *Enterobacter cloacae complex* + Acinetobacter *dijkshoorniae*; PD exit-site positive for *Staphylococcus aureus***						
Administered amount of vancomycin	50 mg/2 L (=25 mg/L)	1000 mg/2 L (=500 mg/L)	50 mg/2 L (=25 mg/L)	50 mg/2 L (=25 mg/L)	50 mg/2 L (=25 mg/L)	
Vancomycin concentration at the end of the dwell (mg/L)	7.1	117.94	10.61	12.36	12.24	
Dialysis solution	icodextrin	glucose (3.86%)	icodextrin	glucose (2.27%)	icodextrin	
*Staphylococcus epidermidis*	1:16	1:16	1:16	1:8	1:8	
*Staphylococcus aureus*	1:8	1:16	1:8	1:8	1:4	
*Enterococcus faecalis*	1:2	1:2	1:2	1:2	1:2	

## Data Availability

The data that support the findings of this study are available from the corresponding author upon reasonable request.

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
