# Peer review of "Intraperitoneally Administered Vancomycin in Patients with Peritoneal Dialysis-Associated Peritonitis: Population Pharmacokinetics and Dosing Implications"

_pharmaceutics, 2023, doi:10.3390/pharmaceutics15051394_

Round 1
Reviewer 1 Report
In everyday medical practice, the use of off-label drugs is common. A very important element is the availability of medical evidence supporting this type of treatment. Important manuscript, results clearly presented, well discussed.
In order to increase the eloquence of the work, I propose to expand several aspects:
- in the introduction, in the first paragraph, there is information about peritoneal dialysis and its effect on pharmacokinetics. I suggest adding information about:
1) unusual routes of drug administration result from clinical necessity, vancomycin is no exception. He can cite examples such as abciximab and intravenous administration, but also intracoronary, intraocular antibiotics in ophthalmology.
2) one paragraph justifying why vancomycin is the purpose of the work (here microbiological and statistical data on infections through the prism of vancomycin use can be cited).
- in table 1 - can ideal body weight/fat-free body mass be provided? can uric acid concentration be added?
- figure 2 - please increase the font size (numbers and description of both axes), increase the size of the points and make the lines bolder.
Author Response
Dear reviewer,
we appreciate the suggestions you made to our manuscript and we accepted them all:
1) unusual routes of drug administration result from clinical necessity, vancomycin is no exception. He can cite examples such as abciximab and intravenous administration, but also intracoronary, intraocular antibiotics in ophthalmology.
As the examples of alternative administration routes, we added a remark on other antibiotics that are administered intraperitoneally to the second paragraph of the introduction as we feel that this is more related to the topic of the article than abciximab and itraocular antibiotics.
2) one paragraph justifying why vancomycin is the purpose of the work (here microbiological and statistical data on infections through the prism of vancomycin use can be cited).
We added a remark on growing occurrence of beta-lactam resistant strains in PD dialysis associated peritonitis to the second paragraph of the introduction. We also mentioned this as the reason why we focused on vancomycin dosing at the last paragraph of the introduction. As this required addition of a new reference (now No. 2) we also renumbered all the subsequent references.
- in table 1 - can ideal body weight/fat-free body mass be provided? can uric acid concentration be added?
Ideal body weight, fat free mass and uric acid levels were added to the table 1.
- figure 2 - please increase the font size (numbers and description of both axes), increase the size of the points and make the lines bolder.
Figure 2 had been redrawn according to the reviewer’s proposal.
Thank you very much for improving the scientific level of our article,
with regards,
JM Hartinger
Reviewer 2 Report
In their manuscript, Jan Miroslav et al. investigated the pharmacokinetics of intraperitoneally administered vancomycin for the treatment of peritonitis patients. The study identified the optimal dosage and administration route for vancomycin in these patients. While the results may be subject to biases related to race and gender, the study provides a promising solution for current antibiotics administration. I have no further comments on the study itself, but suggest that the authors should include the full names of MRSA and AUC in the Keywords section, along with Peritonitis and Vancomycin. Additionally, the authors should acknowledge a limitation of the study in the last paragraph to provide a comprehensive understanding of its potential drawbacks.
Author Response
Dear reviewer,
we appreciate the suggestions you made to our manuscript and we accepted them all:
While the results may be subject to biases related to race and gender, the study provides a promising solution for current antibiotics administration.
We added the information about gender to the table 1; The gender was tested as a covariate for a pharmacokinetic parameters and no significant influence was found. The race remains a possible bias as all but one patient were Caucasians. We added this to the last paragraph of the discussion as a drawback.
I have no further comments on the study itself, but suggest that the authors should include the full names of MRSA and AUC in the Keywords section, along with Peritonitis and Vancomycin.
We added the full wording of MRSA and AUC.
Additionally, the authors should acknowledge a limitation of the study in the last paragraph to provide a comprehensive understanding of its potential drawbacks.
We added a paragraph about the study drawbacks at the end of the discussion.
Thank you very much for improving the scientific level of our article,
with regards,
JM Hartinger
